# On Joint Optimization of UAV-Assisted Covert Communication Systems with NOMA for Hydropower Internet of Things

Zhenchun Le [1], Qing Xu [2], Yining Wang [2], Guowen Hao [3], Weifeng Pan [2], Yanlin Sun [2] and Yuwen Qian [4,*]

1  State Grid Corporation of China, Beijing 100031, China; zhenchun-le@sgcc.com.cn
2  State Grid Electric Power Research Institute, Nanjing 210003, China;
   xuqing@sgepri.sgcc.com.cn (Q.X.); wangyining@sgepri.sgcc.com.cn (Y.W.);
   panweifeng@sgepri.sgcc.com.cn (W.P.); sunyanlin@sgepri.sgcc.com.cn (Y.S.)
3  State Grid Xin Yuan Company Limited, Beijing 100056, China; haoguowen@126.com
4  School of Electronic and Optical Engineering, Nanjing University of Science and Technology,
   Nanjing 210094, China
*  Correspondence: admon@njust.edu.cn; Tel.: +86-189-3603-0253

**Abstract:** The intelligent terminals deployed in hydropower IoT can quickly sense the status of hydropower equipment, thus improving the efficiency of system control and operation. However, communication security between the base station and intelligent terminals challenges the IoT hydropower plant. In this paper, we propose a UAV-assisted covert communication system (CCS), where a UAV acts as the base station to provide communication service to ground terminals monitored by malicious users. To improve access effectiveness, we adopt non-orthogonal multiple access (NOMA) for intelligent terminals to access the hydropower IoT. Since two devices can synchronously access the communication system with the NOMA scheme, we select one terminal to receive covert messages and the other to interfere with the malicious users to detect confidential communications. To maximize the covert rate, we formulate the optimization problem that jointly optimizes the transmit power, the altitude of the UAV, and trajectory under the constraints of covertness and the finite length of the transmission message block. Additionally, we transform the optimization problem into a geometric planning one, which is solved by a developed sequential geometric planning (SGP) approximation algorithm. Simulation results show the proposed algorithm can improve the covert rate compared to the traditional methods.

**Keywords:** information security; hydro-power Internet of Things; covert communication; UAV



## 1. Introduction

Recently, intelligent terminals have become popular in the hydropower Internet of Things (IoT), which can be used to collect boundary data and automatically analyze the collected data, thereby improving the efficiency of hydropower plants. According to [1], there is a scope of study for monitoring the performance under conditions in real-time in hydropower, as it is difficult to predict the behavior of the machine using the existing methods. To improve the real-time and reliability of information exchange of hydropower plants, an IoT-aided hydropower system was proposed in [2] based on message queuing telemetry transport. Furthermore, to enhance energy and spectrum efficiency of the hydropower IoT, the advanced multiple access technology known as non-orthogonal multiple access (NOMA) has been identified as a promising solution for facilitating terminal access to wireless networks due to its advantages, such as low latency, extremely dense coverage, and high fairness [3]. Especially, in hydropower IoT, NOMA can improve energy efficiency and spectrum utilization by providing lower power consumption and more efficient data transmission solutions for IoT devices to support various IoT applications [4]. However, developing a secure communication scheme based on NOMA is a major challenge in IoT networks for hydropower plants.

The production of hydropower plants is characterized by corruptibility and sudden changes, which requires quick and close monitoring of hydropower equipment to ensure safe production. In this case, it is urgent to design the communication system to transmit the monitored data safely, reliably, and regularly to the control center. To ensure absolute security, traditional hydropower plants are not connected to a network or the Internet, and thus bring greater inconvenience and low productive efficiency. However, the amount of sensitive data is very small and can be easily cracked by using cryptographic methods with obtained cipher texts. To improve productive efficiency, covert communication is used for hydropower plants to collect parameters and hand out instructions. In this case, covert communication has attracted much attention as a key technology for network security. Traditionally, covert communication systems (CCS) are designed by using network protocols, such as transmission control protocols (TCP), Internet protocol (IP), and media access control (MAC) protocols [5]. However, these communication systems cannot be used for wireless sensors in IoT systems. In this scenario, wireless covert communication is adopted as an effective method for secure communication of intelligent terminals. However, wireless CCSs cannot achieve a high covert rate in complex environments due to the barriers between the transmitter and receiver.

In a hydropower plant, there are volumes of power generation equipment, hydraulic mechanical equipment, electrical equipment and systems, flood gates, and hydraulic buildings, which form a complex communication environment. Moreover, the IoT network for a hydropower plant usually covers a larger range than traditional IoT networks. Owing to the barriers and the large range coverage, traditional wireless covert communication cannot be deployed in hydropower plants. In this case, the UAV-aided covert communication is used as the base station to receive the collected data from sensors and the handout instructions to the devices. To solve the problem, existing works have focused on secure communication with Unmanned Aerial Vehicle (UAV) communication networks, launching an UAV to avoid being blocked by barriers [6]. Furthermore, the transmission behavior of an UAV cannot be determined, and thus, the UAV-assisted covert communication cannot be easily detected by malicious users [4]. Accordingly, the authors of [7] propose a covert wireless communication method to transmit covert information from Alice to Bob under the monitor of Willie. Moreover, an UAV-assisted millimeter wave wireless system is proposed in [8], where the flight altitude, beam number and transmit power of the UAV are jointly optimized. Similarly, the transmit power and the position of the monitored UAV are jointly optimized in [9] to maximize the covert rate for legitimate receivers on the ground. To ensure covertness, the transmit power and trajectory of the UAV are jointly optimized in [10]. In addition, stationary UAV-aided CCSs are designed in [8,9], where the UAVs hover at a fixed position. However, in these works, the length transmission message block (LTMB) is assumed to be infinite, which is impractical.

In practice, the LTMB transmitted via a channel limits transmission efficiency, which thus requires the transmitted messages with short block lengths to be short for practical CCSs [11]. In this sense, the LTMB affects the performance of a CCS, i.e., the detection performance of Willie, covert rate [12]. For example, according to [12,13], finite LTMB imposes a significant impact on the data rate of the channel from Alice to Bob. Therefore, not only the covertness, but also a specified finite LTMB play important roles in the optimization of the CCS to maximize the covert rate [14]. Since the collected parameters and instruments are small blocks with a few bits for a hydropower plant, we adopt the finite LTMB for the covert channels. To improve the performance of the system, we optimize not only the transmit power of the base station, but also the length of the block length. However, how to optimize the performance of a CCS with a finite LTMB remains a challenge.

Recently, UAV-assisted covert communication with NOMA has been widely investigated. As a typical work, the authors of [15] proposed an optimized scheme of the covert communication of downlink NOMA systems by studying the behavior of the outage probability of NOMA users and the expected minimum detection error probability under the uncertain channel knowledge. Inspired by this idea, the optimization algorithm has been

designed for the intelligent reflecting surface (IRS)-assisted NOMA systems to maximize the covert rates by jointly optimizing the transmit power and the IRS reflect beamforming [16]. Similarly, the covert communication scheme was developed for NOMA systems under uncertain channel distribution information to maximize the expected covert rate [17]. In addition, an iterative algorithm was proposed in [18] for UAV-assisted NOMA communication to jointly optimize time slots, transmit power, and trajectories, which ultimately improves the covert rate of UAV-assisted data transmission. The authors of [19] optimized a multiple inputs multiple outputs (MIMO)-NOMA system by minimizing the delay in UAV-assisted caching networks.

In this paper, we propose an UAV-assisted CCS, in which the UAV provides a covert wireless communication service to terminals accessing hydropower IoT networks via NOMA. The UAV acts as a transmitter of covert messages and thus communicates with terminal devices with line of sight (LoS) without being blocked by the barrier in complex hydropower plants. Furthermore, transmit power and the flying altitude of the UAV are optimized to maximize the covert rate under the constraint of covertness and the length of the transmission block. In particular, the contributions of this paper are listed as follows.

- To protect privacy in the hydropower IoT, we propose an UAV-assisted covert communication scheme for the hydropower IoT, where the UAV can communicate with terminals via covert communication channels. Furthermore, the edge terminals the hydropower IoT by using NOMA to reduce energy and spectrum consumption. Moreover, the proposed UAV-assisted covert communication scheme can also be applied to those IoT systems with complex communication environments and large cover ranges.
- We propose a UAV-aided CCS by using NOMA, where the UAV devices act as a transmitter to transmit covert messages under the monitor of malicious users, one accessed terminal with NOMA receives covert messages, and the other accessed terminal jams the communication of the eavesdropper.
- To maximize the covert rate, we formulate the optimization problem to optimize the transmit power and the altitude of the UAV under a finite transmission block and the covertness.
- To solve the formulated nonconvex optimization problem, we first convert the signal programming (SP) into a geometric planning (GP) problem and then adopt a symbolic geometric planning approximation (SGPA) algorithm to solve the optimization problem. Furthermore, the convergence is proved by verifying the given conditions.

The remainder of the paper is organized as follows. In Section 2, we present the UAV-assisted CCS with NOMA and introduce covertness as the measurement metric of the CCS. In Section 3, the optimization problem of the CCS is formulated by optimizing the finite block length under the covertness constraint. Then, the joint optimization problem is transformed into a SP problem. The solution algorithm with the approximation of SGPA is developed, and the convergence of the algorithm is analyzed. Section 4 presents the numerical results, and the conclusion is given in Section 5.

## 2. System Model

Figure 1 demonstrates the UAV-assisted CCS deployed in a hydropower IoT, where the UAV serves as the transmitter of covert messages, providing covert communication services to terminal devices. According to [20,21], NOMA can be used for multi-connectivity, high-speed, and high-reliability scenarios in 5G communication networks to improve performance and efficiency, which can support two devices synchronously accessing the IoT in a hydropower plant. Therefore, in the UAV-assisted CCS with NOMA, two cooperative users, denoted as U1 and U2, receive the covert messages transmitted from the UAV. In addition, there is a monitor denoted as Willie, who keeps monitoring the communication between the UAV and legitimate users to detect covert communications.

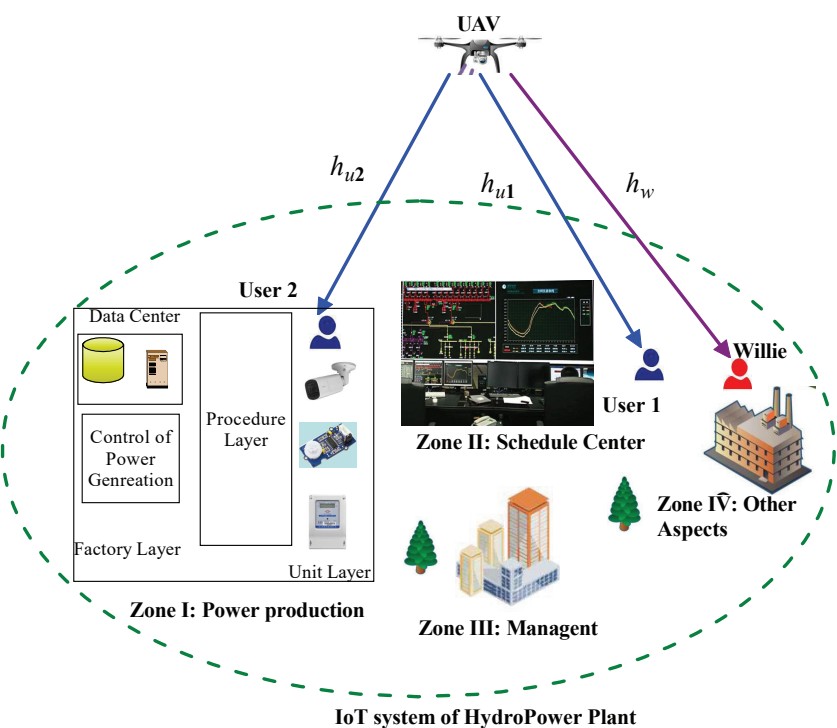

**Figure 1.** UAV-assisted covert communication system deployed in a hydropower plant consisting of several zones, denoted as Zone I, Zone II, Zone III, and Zone IV, where Zone I is used for production, Zone II for schedule, Zone III for management, and Zone IV for other aspects.

In this paper, we adopt a multiuser-based NOMA power allocation method that can utilize different channel gains among multiplexed users. First, we use the NOMA technique to send signals from different users. On the receiver side, successive interference cancellation (SIC) is used to detect the desired signal. Second, the power allocation of the user depends on the distance between the user and the UAV, which means the user with a far distance receives more transmit power than the user with a normal distance. Then, based on the above power allocation method, the power of user one and user two is jointly optimized. In addition, we use the finite LTMB for the UAV to transmit covert messages to the terminals, which is assumed to be *N*. Since NOMA is adopted as the access mode for terminals, *N* sub-channels are used by the UAV to transmit covert messages. Add to that, we assume that the UAV knows Willie's position.

Without loss of generality, we assume that U1 is located far from the UAV and close to Willie, called weak NOMA user, and that U2 is close to the UAV and far from Willie, called strong NOMA user. Since the rate of the channel between the UAV and U2 is higher than the rate between UAV and U1, the transmit power of the UAV of the signal to the U2 is lower than that of U1. In this scenario, the UAV transmits the covert messages to strong NOMA users, i.e., U2, and U1 is selected as a jammer to jam Willie to detect the covert communication between the UAV and U2. In hydro-power IoT, U1, U2, and Willie are sensors or devices, equipped with a single antenna.

### 2.1. Covert Wireless Communication with UAV

In this paper, we adopt the Cartesian coordinate system to describe the position, where the coordinates of U1, U2, and Willie are $q_j = [x_j, y_j], j = u_1, u_2, w$. In addition, the coordinates of the best position for UAV communication are $q_i = [x_i, y_i]$, and the altitude of UAV is denoted by $H$, $H_{\min} \le H \le H_{\max}$. Let $h_j, j \in \{u_1, u_2, w\}$ be the channel fading from the UAV to the users, where $u_1$, $u_2$, and $w$ denote U1, U2, and Willie, respectively.

The air-to-ground channel model between the UAV and the users is provided by 3GPP specifications. Accordingly, the path loss depends on the line-of-sight (LOS) and non-line-

of-sight (NLoS) links. For low-altitude UAVs and short-range LoS communication, the UAV-ground channels are mostly been dominated by the LoS link. For example, the authors of [22] proposed a secure communication scheme for the NOMA-based UAV-MEC system towards a flying eavesdropper. The channels are assumed to be well modeled by the quasi-static block fading LoS links and follow the distance-dependent path loss model. In addition, ref [23] proposed a UAV-assisted multi-user communication system based on NOMA, which uses the line of sight air-to-ground communication. Therefore, according to [18,24], the communication between the UAV and a ground user is modeled as an LoS link, and thus we can obtain the expression

$$h_j = \sqrt{\frac{\beta}{\left\| q_i - q_j \right\|^2 + H^2}}, j = u_1, u_2, w \tag{1}$$

Since $h_w$ is known to Willie, as additional power is received from the UAV, Willie can determine that covert communications exist. To avoid being detected by Willie, the transmit power of the UAV is random. For the communication between UAV and U1, a random power transmission scheme is adopted, where the transmit power follows a uniform distribution over $\left[0, P_{u_1}^{\max}\right]$. Accordingly, the probability density function of $P_{u_1}$ is given in Equation (2)

$$f_{P_{u_1}}(x) = \begin{cases} \frac{1}{P_{u_1}^{\max}}, & \text{if } 0 \le x \le P_{u_1}^{\max}, \\ 0, & \text{otherwise}, \end{cases} \tag{2}$$

where $P_{u_1}^{\max}$ is the maximum transmit power of the UAV to U1.

When the UAV transmits signals to U1 and U2, the received signals can be expressed as

$$\mathbf{y}_{u_1}[k] = \sqrt{P_{u_1}} h_{u_1} \mathbf{x}_{u_1}[k] + \sqrt{P_{u_2}} h_{u_1} \mathbf{x}_{u_2}[k] + \mathbf{n}_{u_1}[k], \tag{3}$$

and

$$\mathbf{y}_{u_2}[k] = \sqrt{P_{u_1}} h_{u_2} \mathbf{x}_{u_1}[k] + \sqrt{P_{u_2}} h_{u_2} \mathbf{x}_{u_2}[k] + \mathbf{n}_{u_2}[k], \tag{4}$$

where $k = 1, 2, \ldots, N$, $x[k]$ is denoted as the signal transmitted by the UAV in the $k$-th covert channel. $P_{u_1}$ and $P_{u_2}$ denote the transmit power of the UAV communicating with U1 and U2, respectively. $\mathbf{n}[k]$ is the received Gaussian white noise with the variance, denoted by $\sigma_{u_1}^2$, being 0. In the proposed CCS with NOMA, the transmission between the UAV and U1 can provide a cover for the covert transmission of U2. Since $P_{u_1} > P_{u_2}$, by using the SIC decoder, we can express the received signal-to-interference plus noise ratio (SINR) of U1 as

$$\gamma_{u_1} = \frac{P_{u_1} h_{u_1}^2}{P_{u_1} h_{u_1}^2 + \sigma_{u_1}^2}. \tag{5}$$

The received SINR of U2 is

$$\gamma_{u_2} = \frac{P_{u_2} h_{u_2}^2}{\sigma_{u_2}^2}. \tag{6}$$

Then, with the obtained channel model, we can analyze the performance of the system with finite LTMB.

### 2.2. Performance of the System with Finite LTMB

According to [25], the decoding error probability (DEP) at U2 cannot be negligible when $n$ is finite. Given a DEP, denoted by $\delta$, according to [25,26], the data rate from UAV to U2 can be approximated as

$$R \approx \log_2(1 + \gamma_{u_2}) - \sqrt{\frac{\gamma_{u_2}(\gamma_{u_2} + 2)}{n(\gamma_{u_2} + 1)^2}} \frac{Q^{-1}(\delta)}{\ln(2)} + \frac{\log_2(n)}{2n}, \tag{7}$$

where $Q^{-1}(\cdot)$ is the inverse of the $Q$ function.

Similarly, given the data rate $R$, the DEP at U2 is given by

$$\delta = Q\left(\frac{\sqrt{n}(1+\gamma_{u_2})\left(\ln(1+\gamma_{u_2})+\frac{1}{2}\ln(n)-R\ln 2\right)}{\sqrt{\gamma_{u_2}(\gamma_{u_2}+2)}}\right). \tag{8}$$

*2.3. Detection of Covert Communications by Willie*

In this paper, we assumed that Willie knows the transmit power of the UAV, the channel gain of the UAV to Willie, and the noise distribution at Willie. In this case, the signal received by Willie can be expressed as a binary hypothesis test

$$y_w[i] = \begin{cases} \sqrt{P_{u_1}}h_w x_1[i]+n_w[i], & \mathcal{H}_0 \\ \left(\sqrt{P_{u_1}}x_1[i]+\sqrt{P_{u_2}}x_2[i]\right)h_w+n_w[i], & \mathcal{H}_1 \end{cases}, \tag{9}$$

where $\mathcal{H}_0$ is the hypothesis that the UAV is not communicating with the covert user U2 and the hypothesis that the UAV is communicating with U2. $n_w[i]$ is Willie's received Gaussian noise with the variance 0. According to [9,27,28], the optimal decision criterion for Willie's total detection error probability is

$$T_w \triangleq \frac{1}{N}\sum_{i=1}^{N}|y_w[i]|^2 \underset{\mathcal{D}_0}{\overset{\mathcal{D}_1}{\underset{<}{\gtrless}}} \tau, \tag{10}$$

where $T_w$ denotes the received power of Willie, $\tau$ is defined as Willie's detection threshold, and $D_0$ and $D_1$ are the binary decisions supporting $\mathcal{H}_0$ and $\mathcal{H}_1$, respectively.

Note that the transmit power between the UAV and the user is considered to be fixed within each time slot. However, the transmit power of the UAV is assumed to change in different time slots for the reason of being undetectable.

Assuming that the channels are infinitely many, i.e., $N \to 0$, combining Equations (6) and (7), we can derive the following expression:

$$T_w = \begin{cases} P_{u_1}|h_w|^2+\sigma_w^2, & \mathcal{H}_0 \\ (P_{u_1}+P_{u_2})|h_w|^2+\sigma_w^2, & \mathcal{H}_1 \end{cases}, \tag{11}$$

Assuming that the initial probabilities of $\mathcal{H}_0$ and $\mathcal{H}_1$ are equal, Willie's detection performance is measured by its DEP, which is defined as

$$\xi = P_F + P_M, \tag{12}$$

where $P_F \triangleq \Pr\{\mathcal{D}_1 \mid \mathcal{H}_0\}$ is the false alarm probability (FAP) and $P_M \triangleq \Pr\{\mathcal{D}_0 \mid \mathcal{H}_1\}$ is the missed detection probability (MDP).

According to Equations (10) and (11), we can derive the total detection error probability $\xi$, which contains an incomplete gamma function [29]. In this paper, we use the Kullback–Leibler (KL) scatter as a lower bound $\xi^*$ for the minimum detection error rate, expressed as

$$\xi^* \geq 1 - \sqrt{\frac{1}{2}\mathcal{D}(P_F, P_M)}, \tag{13}$$

where $\xi^* \geq 1 - \varepsilon$ is defined as the covertness constraint and $\varepsilon$ is the covertness constraint limiting constant. According to Equation (13), it holds that

$$\mathcal{D}(P_F, P_M) \leq 2\varepsilon^2. \tag{14}$$

In the following sections, Equation (14) is used as the covertness constraint.

### 3. Problem Optimization

*3.1. Problem Formulation*

From the square root law theorem, the covert rate is no more than $\mathcal{O}(\sqrt{n})$ bits for the CCS with $n$ covert channels, when $n \to \infty$. For the CCS with finite channels, we cannot use the square root theorem to calculate the covert rate, since the DEP of the system with finite covert channels cannot be negligent [30,31]. In this paper, the covert rate of the CCS is considered as the objective function of the CCS with finite LTMB. Thus, we can obtain the following expression:

$$\eta = nR(1 - \delta), \tag{15}$$

where $\eta$ is the average number of bits transmitted from UAV to U2 with the $n$ covert channel.

To maximize the covert rate, we formulate the joint optimization problem:

$$(P1) \max_{n, P_{u1}, P_{u2}, H} \eta, \tag{16}$$

$$s.t.$$

$$n \leq N, \tag{17}$$

$$P_{u_1} + P_{u_2} \leq P_{\max}, \tag{18}$$

$$P_{u_1} > P_{u_2}, \tag{19}$$

$$\mathcal{D}(P_F, P_M) \leq 2\varepsilon^2, \tag{20}$$

$$H_{\min} \leq H \leq H_{\max}. \tag{21}$$

where $P_{\max}$ is the maximum transmit power of the UAV. To maximize the covert rate, we optimize the UAV-assisted NOMA CCS under several constraints, including LTMB, maximum transmit power of the UAV, covertness, and flying altitude. This is because the design of $n$ and $P$ affects both the covert rate of the channel from UAV to U2 and the detection performance of Willie. Also, due to the mobility of the UAV, the flight altitude constraint is adopted to maximize the covert with guaranteed covertness. In the following, we first find the optimal values of $n$ and $P$ that maximize the effective throughput $\eta$ under the $\mathcal{D}(P_F, P_M) \leq 2\varepsilon^2$ and $n \leq N$ constraints. Secondly, due to the nonconvexity of problem P1, we need to focus on converting the covertness constraint Equation (20). Finally, we design the SGPA algorithm.

*3.2. Optimization of Transmit Power*

In this subsection, we find the value of $n$ ($n \leq N$) to maximize the covert rate under the constraint as follows:

$$\mathcal{D}(P_F, P_M) \leq 2\varepsilon^2, \tag{22}$$

Assuming that the UAV communicates with U2, the KL scatter $\hat{\mathcal{D}}(P_F, P_M)$ can be expressed as

$$\hat{\mathcal{D}}(P_F, P_M) = n \left[ \ln\left( \frac{P + \sigma_w^2}{\sigma_w^2} \right) - \frac{P}{P + \sigma_w^2} \right]. \tag{23}$$

According to the limitation of the covert channel numbers, the optimal $\mathcal{D}(P_F, P_M)$ and $n$ can be assumed as

$$n^* = N, \tag{24}$$

and

$$\hat{\mathcal{D}}(P_F, P_M) = n^* \left[ \ln\left( \frac{P + \sigma_w^2}{\sigma_w^2} \right) - \frac{P}{P + \sigma_w^2} \right] = 2\varepsilon^2. \tag{25}$$

where $n^*$ is the optimal value.

According to [32], we can obtain the optimal solution of transmit power:

$$P^* = \left( \sigma_w^2 + P^* \right) \left[ \ln\left( \frac{P^*}{\sigma_w^2} + 1 \right) - 2\varepsilon^2 N \right]. \tag{26}$$

In Equation (26), as $n = N$, the covert rate achieves the maximum value. From Equation (26), we can see that the $P^*$ decreases as $n$ increases. Traditionally, to ensure the covertness, we select PU1 and PU2, which are smaller than $P_{\max}$, rather than $P_{\max}$ as adopted in traditional systems. With the over-KL scatter $\mathcal{D}(P_F, P_M)$ can be rewritten as

$$\mathcal{D}(P_F, P_M) = N\left[\ln\left(1 + \frac{P_{u_2}^* h_w^2}{P_{u_1} h_w^2 + \sigma_w^2}\right) - \frac{P_{u_2}^* h_w^2}{(P_{u_2}^* + P_{u_1})h_w^2 + \sigma_w^2}\right]. \tag{27}$$

$P_{u_2}^*$ is analogous to $P^*$, and the same level of covert performance can be achieved by optimizing $P_{u_2}^*$ when $n^* = N$ is optimal.

### 3.3. Covertness Constraint

For the optimal finite block, problem $P1$ can be transformed as

$$(P2) \max_{P_{u_1}, P_{u_2}^*, H} \eta^*, \tag{28}$$

$s.t.$

$$P_{u_1} + P_{u_2}^* \le P_{\max}, \tag{29}$$

$$P_{u_1} > P_{u_2}^*, \tag{30}$$

$$\mathcal{D}(P_F, P_M) \le 2\varepsilon^2, \tag{31}$$

$$H_{\min} \le H \le H_{\max}. \tag{32}$$

However, $(P2)$ is still not convex. In the following, we first cope with the constraint of covertness given in Equation (19), which can be expressed with the KL scatter $\mathcal{D}(P_F, P_M)$, see Equation (33).

$$\begin{aligned}\mathcal{D}(P_F, P_M) &= -N\ln\left(1 - \frac{P_{u_2} h_w^2}{(P_{u_2} + P_{u_1})h_w^2 + \sigma_w^2}\right) - \frac{N P_{u_2} h_w^2}{(P_{u_2} + P_{u_1})h_w^2 + \sigma_w^2} \\ &= Nf(t),\end{aligned} \tag{33}$$

where $f(t)$ and $t$ are, respectively, defined as

$$f(t) = -\ln(1-t) - t \tag{34}$$

and

$$t = \frac{P_{u_2} h_w^2}{(P_{u_2} + P_{u_1})h_w^2 + \sigma_w^2} \tag{35}$$

Since $f(0) = 0$ and $\lim_{t\to 1} f(t) = \infty$, it holds that $f(t)$ is a monotonically increasing function.

Combining Equations (20) and (33) yields

$$f(t) \le \frac{2\varepsilon^2}{N}. \tag{36}$$

Let $t^*$ be the root of $f(t) = 2\varepsilon^2/N$, $t \in (0,1)$, and substituting Equation (1) into Equation (34), Equation (36) can be expressed as

$$\frac{P_{u_2}\beta}{(P_{u_2} + P_{u_1})\beta + \sigma_w^2\left(\|q_i - q_w\|^2 + H^2\right)} \le t^*. \tag{37}$$

In this way, Equation (20) can be transformed into a symbolic function, see Equation (37). With the standard form of the symbolic geometric programming (SGP) problem [33], we transform the maximization problem in $(P2)$ into a minimization problem:

$$(P3) \max_{P_{u_1}, P_{u_2}^*, H} \frac{1}{\eta^*}, \tag{38}$$

s.t.

$$P_{u_1} + P_{u_2}^* \leq P_{\max}, \tag{39}$$

$$P_{u_1} > P_{u_2}^*, \tag{40}$$

$$\frac{P_{u_2}\beta}{(P_{u_2} + P_{u_1})\beta + \sigma_w^2\left(\|q_i - q_w\|^2 + H^2\right)} \leq t^*, \tag{41}$$

$$H_{\min} \leq H \leq H_{\max}. \tag{42}$$

However, the symbolic geometric programming of $(P3)$ is a class of nonlinearly constrained optimization problems that cannot be solved directly. In this context, we employ Continuous Geometric Planning to approximate $(P3)$.

### 3.4. Approximation with Continuous Geometric Planning

To solve $(P3)$, we change the left-hand side of Equation (37) into a positive term form in the GP. According to [25], the positive denominator in Equation (37) is then replaced by using a single term with the geometric mean approximation.

First, the denominator of the positive term is denoted as

$$\theta = (P_{u_1} + P_{u_2})\beta + \sigma_w^2\left(\|q_i - q_w\|^2 + H^2\right), \tag{43}$$

Then, the approximation of the positive term can be expressed as

$$\tilde{\theta} = \left(\frac{\sigma_w^2\|q_i - q_w\|^2}{\lambda_0}\right)^{\lambda_0} \times \left(\frac{P_{u_2}\beta}{\lambda_1}\right)^{\lambda_1} \times \left(\frac{P_{u_1}\beta}{\lambda_2}\right)^{\lambda_2} \times \left(\frac{\sigma_w^2 H^2}{\lambda_3}\right)^{\lambda_3}. \tag{44}$$

Define $A_0 = \sigma_w^2\|q_i - q_w\|^2$, $A_1 = P_{u_2}\beta$, $A_2 = P_{u_1}\beta$, $A_3 = \sigma_w^2 H^2$, and we can derive

$$\lambda_i = \frac{A_i}{\theta}, i = 0, \cdots, 3. \tag{45}$$

Therefore, this method is derived from the geometric mean inequality of classical arithmetic [33] by replacing the arithmetic mean with the geometric mean. In this paper, the SP problem approximates a standard GP problem, and then $(P3)$ can be approximated with the GP:

$$(P4) \max_{P_{u_1}, P_{u_2}^*, H} \frac{1}{\eta^*}, \tag{46}$$

s.t.

$$\frac{P_{u_2}\beta}{\tilde{\theta}} \leq t^*, \tag{47}$$

$$P_{u_1} + P_{u_2}^* \leq P_{\max}, \tag{48}$$

$$P_{u_1} > P_{u_2}^*, \tag{49}$$

$$H_{\min} \leq H \leq H_{\max}. \tag{50}$$

Accordingly, the problem $P4$ is convex and can be solved with CVX.

### 3.5. Optimization Algorithm

With the approximation method, we develop the SGPA, as presented in Algorithm 1.

In Algorithm 1, we first set the initialization values for various parameters. Then, on line 1, we set a feasible value for the altitude of the UAV, the length of the finite block, and the transmit power, i.e., $N$, $H^0$, $P_{u_1}^0$, $P_{u_2}^{*0}$). To ensure the feasibility, the height $H^0$ is set to its minimum value, i.e., $H_{\min}$. According to the finite block length $n^* = N$, the transmit power of U2 is set to a small positive value $P_{u_2}^{*0}$, which can guarantee covertness. On the contrary, the transmit power of U1 is set to be as high as possible, which can be expressed as $P_{\max} - P_{u_2}^{*0}$). On line 3, The approximation $\lambda_i$ is updated according to Equations (44) and (45). Then, the approximate solution is derived by the GP problem, if the optimization result of the throughput satisfies the threshold $\alpha$. If the accuracy is not satisfied, jump to step 3 to continue the cycle. Finally, the solution to the problem is obtained, and the optimized parameters are outputted.

---

**Algorithm 1** Iterative optimization algorithm.

---

**Input:** Set the initial values for $n$, $\varepsilon$, $P_{\max}$, $H_{\min}$, $H_{\max}$, and $\beta$.
**Output:** $H = H^{r+1}, P_{u_2}^* = P_{u_2}^{r+1} P_{u_1} = P_{u_1}^{r+1}, \eta = \eta^{r+1}$ as the optimal solution.

1: Find a feasible solution by setting $N, H^0, P_{u_1}^0, P_{u_2}^{*0}$.
2: Set the number of iterations r = 0.
3: Update $\lambda_i, (i = 0, \ldots, 3)$ .
4: Obtain the approximate solution of the GPB $H^{r+1}, P_{u_1}^{r+1}, P_{u_2}^{r+1}, \eta^{r+1}$.
5: **if** $\left\| \eta^{r+1} - \eta^r \right\| \leq \alpha$ **then**
　　break.
6: **else**
　　$r = r + 1$.
7: **end if**
8: return to step 2.

---

By comparing the size analysis of $\tilde{\theta}$ and $\theta$, we can observe that the geometric mean approximation sequence converged to the point, at which the Karush–Kuhn–Tucker (KKT) condition is satisfied.

According to [33], we list these conditions as follows. First, $\theta \leq \tilde{\theta}$. This is a tight constraint given in Equation (37) to ensure that any solution to the GP problem is a feasible solution to the SP problem. Since the geometric mean method is used to approximate the arithmetic mean, this condition can be guaranteed.

Second, $\theta\left(H^r, P_{u_2}^r, P_{u_1}^r\right) \leq \tilde{\theta}\left(H^r, P_{u_2}^r, P_{u_1}^r\right)$, with which we can ensure that the solution to the GP problem can reduce value for the objective function in each iteration. From Equations (44) and (45), the geometric mean is equal to the arithmetic mean, and thus the condition can be satisfied.

Third, $\nabla\theta\left(H^r, P_{u_2}^r, P_{u_1}^r\right) \leq \nabla\tilde{\theta}\left(H^r, P_{u_2}^r, P_{u_1}^r\right)$, with which we can ensure that the KKT condition for the SP problem can be satisfied by the convergence of a series of GP approximations.

## 4. Numerical Results

In this section, to evaluate the performance of the proposed algorithm, we constructed the simulation system according to a hydropower plant with a complex communication environment, as illustrated in Figure 1. Specifically, 2000 sensors are used to collect data from hydropower equipment, and 4000 devices are used to receive instructions from the UAV via the covert communication channel in the hydropower plant. We conduct simulations on Matlab2017a and present the numerical results for the Intelligent UAV-assisted CCS with NOMA, which is applied in the hydropower IoT. To evaluate the performance of the proposed system, we implement finite length of transmission message block-Time Division Multiple Access (FLTMB-TDMA) [34], and infinite length of transmission block as the benchmarks [35]. Furthermore, Algorithm 1 is designed based on Successive Con-

vex Approximation (SCA) and Dinkelbach to solve the formulated optimization problem. The parameters are listed in Table 1.

**Table 1.** Simulation parameters.

| Parameter | Meaning | Value |
|:---:|:---:|:---:|
| $T$ | Number of time slots | 30 |
| $L$ | Duration of time slot | 1 s |
| $H$ | Altitude of UAV | 50 m |
| $V_{max}$ | Maximum speed of UAV | 20 m/s |
| $d_t$ | Duration of the time slot | 1 s |
| $P$ | Transmission power | 20 dB |
| $\sigma^2$ | Power of noise | 60 dB |
| $\epsilon$ | Covertness measurement | $10^{-5}$ |
| $\delta$ | Detection error probability of UAV | $10^{-2}$ |

Figure 2 shows the UAV trajectories of the proposed scheme, infinite length of transmission message block NOMA (IFLTMB-NOMA), and finite length of transmission message block NOMA (FLTMB-TDMA) with periods $T = 45$ s.

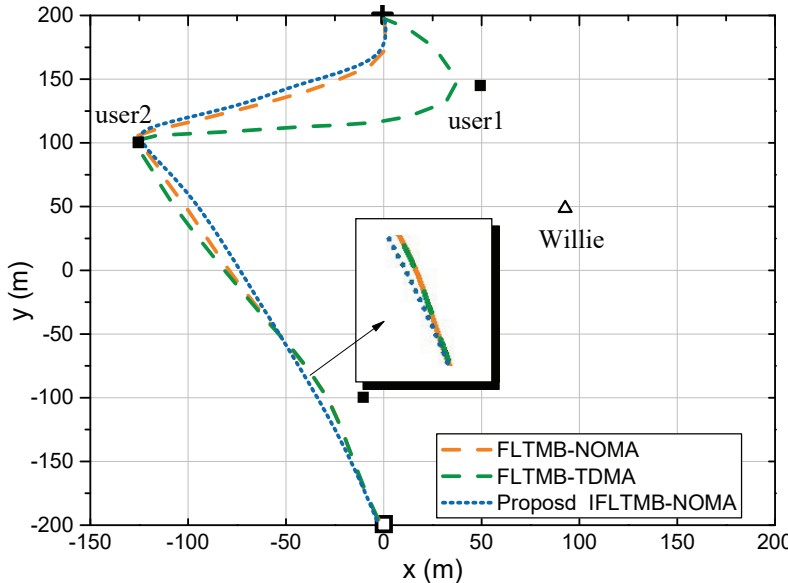

**Figure 2.** The trajectories of UAV for different accessing schemes, i.e., FLTMB-NOMA, FLTMB-TDMA, and IFLTMB-NOMA.

From Figure 2, we can observe that when the flight period is small, the trajectories of the UAV adopting three schemes are much closer. There is also a small difference among the three schemes, the IFLTMB-NOMA scheme is more sensitive to the eavesdropper than the FLTMB-TDMA scheme when the UAV flies close to U1. When FLTMB-TDMA is adopted, the UAV flies closer to the user in the presence of Willie. In the TDMA scheme, due to the limited flight time, the time slots are first allocated to the closest user of the UAV to achieve a better channel quality and thus improve the data rate. In this way, the trajectory of the UAV with the TDMA scheme is closer to Willie than that of the NOMA scheme. We also observe that the UAV using the NOMA scheme is farther from Willie, which ensures the covertness of the covert communication.

Figure 3 plots the covert rate of the proposed UAV aided with NOMA versus iterative numbers. In general, the global optimum solution can usually be obtained.

We observe that the proposed SGPA algorithm can converge quickly with different parameter settings. When the number of SGPA iterations reaches around three, the covert

rate of the proposed CCS remains almost constant with the increase in iteration numbers. In addition, according to [33], the proposed algorithm converges polynomial approximation at a locally optimal point.

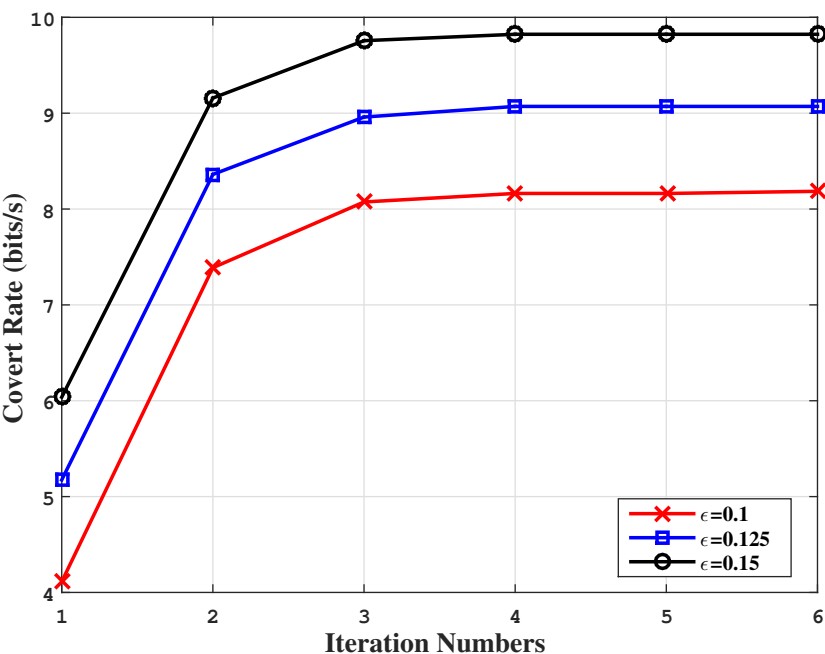

**Figure 3.** Achieved covert rate of the proposed UAV-assisted CCS with NOMA versus the number of iterations for different detection error rates.

Figure 4 shows the covert rate of the proposed UAV-assisted CCS with NOMA by applying the joint optimization method on the flight altitude of the UAV and the transmit power versus the flying altitude of the UAV.

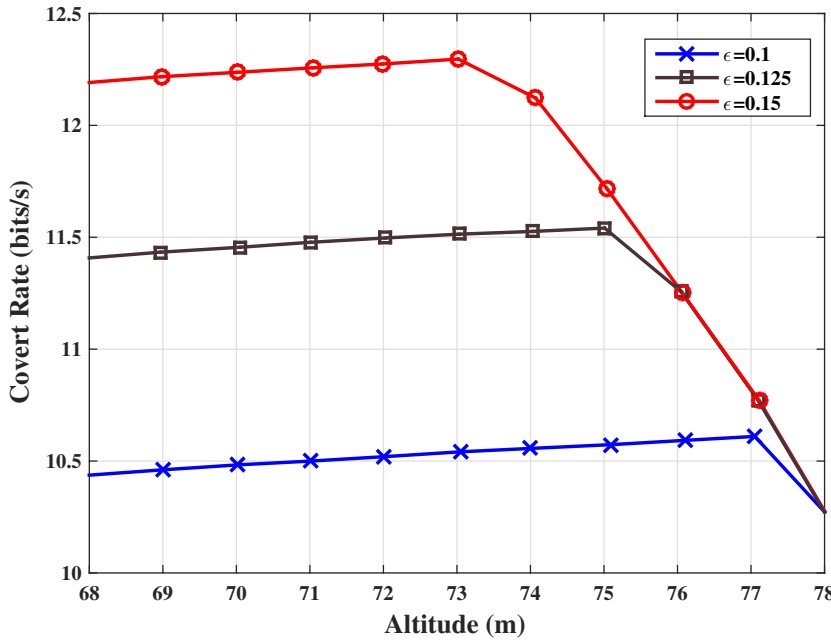

**Figure 4.** The covert rate of UAV-assisted CCS with NOMA under joint optimization of different UAV flight altitudes with different detection error rates.

From Figure 4, we can observe that the covert rate first obtains a slow increase as $H$ increases and then decreases rapidly when the altitude of the UAV is larger than 73 m.

The reason is that the barrier between the UAV and the receiver can no longer affect covert communication, which leads to a slow increase in the covert rate. Second, we can observe that the detection rate of covert communication of Willie can significantly limit the performance of covert communication. For example, the covert rate can achieve a 10% increase when the detection rate varies from 0.1 to 0.125 and from 0.125 to 0.15. On the other hand, when $H$ is large enough beyond a threshold, the covert decreases quickly. This is due to the fact that as $H$ is larger than the threshold, reliable covert communication between the UAV and the receiver cannot be guaranteed owing to the limitation of the transmit power. In addition, when the UAV is far away from Willie beyond a certain threshold, the detection probability decreases, leading to the constraint formulated in Equation (38) no longer playing a role. In this case, the transmit power decreases, which also decreases the covert rate.

Figure 5 demonstrates the covert rate of the proposed UAV-assisted CCS with NOMA versus different finite LTMB with different noise variances and detection rates for each per channel, denoted as $\eta/N$.

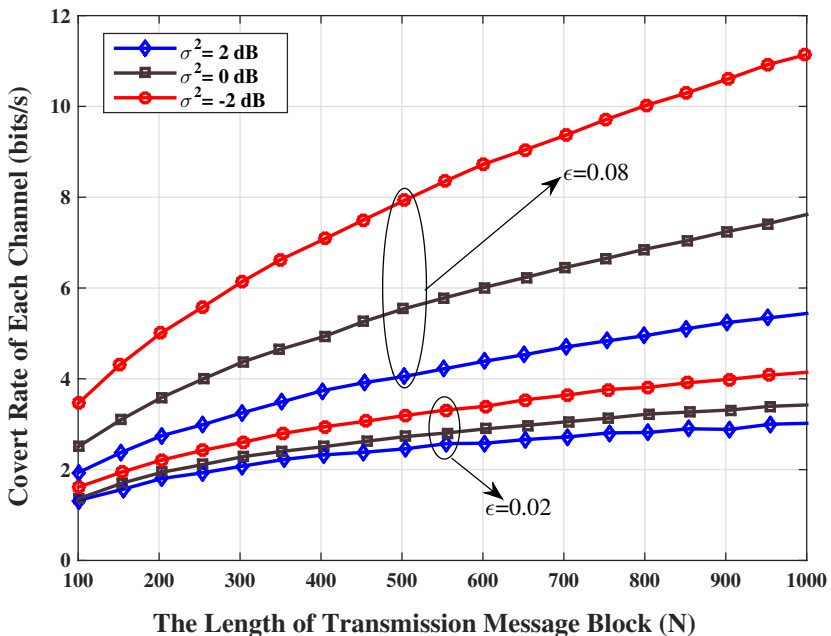

**Figure 5.** The covert rate of UAV-assisted CCS with NOMA versus detection rate for different finite block lengths, different noise variances, and per channel.

From Figure 5, we first observe that as N increases, $\eta/N$ also increases smoothly, which is caused by the optimization of finite blocks and the transmit power, as described in Section 3. When the $\eta$ is optimal, and covertness is satisfied, the maximum value of the finite block transmission channel is $N$. Second, when varying the value of the covertness constraint constant $\varepsilon$, there is a significant change in $\eta/N$ with a slight increase in $\varepsilon$. Therefore, at the same height, the covert rate increases with $\varepsilon$, which indicates that the maximum achievable rate is sensitive to the covertness requirement. Third, we compare the covert rate versus three different noise variances, where the noise variance is the variance value of the U2. In this simulation, the variance value at Willie is set as $\sigma_w^2 = 1$. We observe that the smaller the noise variance of the U2 the larger the covert channel of each channel can be achieved. And this phenomenon increases with the increase in the $\eta/N$ variance at different $\sigma^2$.

Figure 6 plots the covert rate $\eta$ when $\xi^* \geq 1 - \varepsilon$ is satisfied against the decoding error probability $\delta$.

In Figure 6, we first observe that $\eta$ reaches a maximum value when $\delta$ increases up to a certain value, which can help us to find the optimal value of $\delta$. After obtaining the optimal

value of $\delta$, the covert decreases slowly as the detection rate of Willie keeps increasing. Second, the maximum throughput becomes larger as the finite block length increases with the same decoding error probability, which proves the result shown in Figure 5.

In Figure 7, the performance is compared to the proposed SGPA algorithm with two benchmark schemes. One benchmark is the fixed altitude optimization scheme (FH), where the hover height is set to the lowest altitude (i.e., $H = H_{\min}$) [36]. The second benchmark is the fixed power optimization scheme (FP), where the UAV is launched at the maximum transmit power (i.e., $P_{u_1} + P_{u_2} = P_{\max}$) [10].

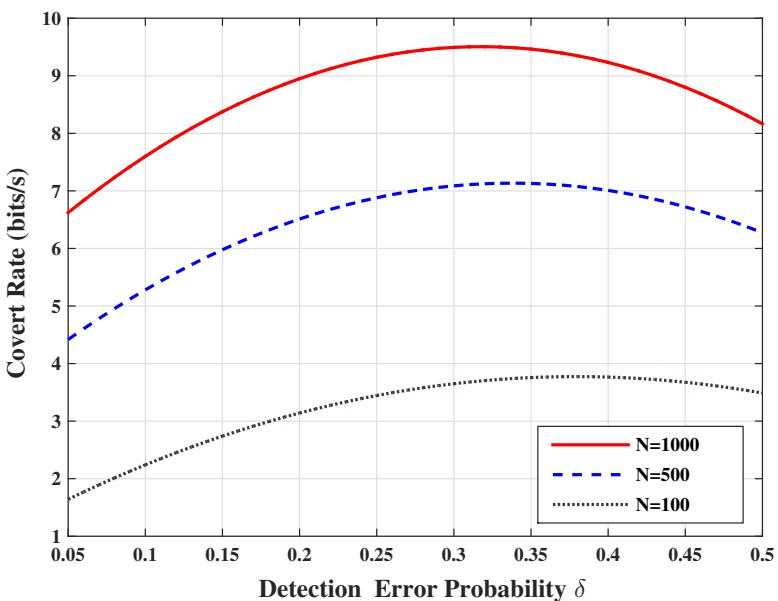

**Figure 6.** The covert rate $\eta$ for different decoding error probabilities $\delta$ when $\xi^* \geq 1 - \varepsilon$.

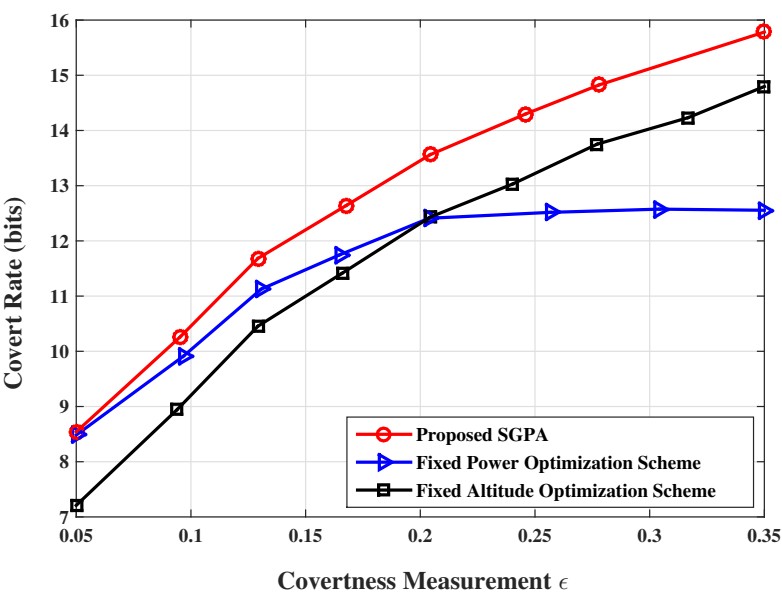

**Figure 7.** The comparison between the performance of the SGPA algorithm and two benchmark schemes, i.e., FH and FP.

From Figure 7, we can observe that $\eta$ increases with $\varepsilon$ for all three schemes. However, the covert rate $\eta$ does not increase with $\varepsilon$ in the FP when $\varepsilon$ is larger than 0.2, tending to be flat. The gradual increase in FP is due to the gradual relaxation of the covertness constraint. For the FP scheme, the covert tends to be a constant, since Willie's detection no longer

affects the covert communication when the $\varepsilon$ is large enough and the optimal flying altitude of the UAV reaches $H_{\min}$. By comparison, the proposed SGPA scheme can obtain a higher covert rate than the benchmark schemes, which verifies the outperforming of the joint optimization of the SGPA algorithm. In addition, we observe that the $\varepsilon$ is larger, and FH can obtain better performance than FP, due to the relaxation of the covertness constraint.

## 5. Conclusions

In this paper, we have proposed a UAV-assisted CCS by using NOMA for multiple users in hydropower IoT systems, where the UAV can provide NOMA services to two users at the same time. One user is selected as the receiver of the covert messages, and the other acts as the jammer used to interfere with the detection of the covert communications. Besides the hydropower IoT network, the proposed UAV-assisted communication scheme can be deployed in other complex IoT networks with large coverage areas. To improve the performance of covert communications, the UAV is used as the transmitter of covert messages due to its mobility and the LOS link between the UAV and the served users. To maximize the covert rate, we formulated a jointly optimized problem to optimize the flying altitude of the UAV, the transmit power, and the block length of covert communication channels. It is first demonstrated that the covert rate can be maximized when all available channels are utilized by setting the maximum finite block length as its maximum value. To solve the proposed optimization problem, we first convert the optimization problem to an SP one and then develop an SGPA-based algorithm solved by the problem. Simulation results show that the proposed CCS can achieve a positive covert rate, and the developed algorithm can fast convergence in comparison with traditional optimization algorithms.

This work serves as a first foray into the design of UAV-assisted covert communication with NOMA. Many interesting directions follow this work and deserve further investigation. First, covert wireless communication aided by UAVs can serve multiple users. How to extend the traditional covert wireless communication model to that with more receivers of covert messages and more wardens remains open. Second, since multiple antennas are equipped in the base station, how to select more than one antenna from all antennas to transmit covert messages deserves further investigation.

**Author Contributions:** Conceptualization, Y.W. and Y.S.; methodology, Z.L.; software, Q.X.; validation, Z.L., Q.X. and Y.Q.; formal analysis, Z.L.; investigation, Y.Q.; resources, G.H.; data curation, Q.X.; writing—original draft preparation, Q.X.; writing—review and editing, Z.L.; visualization, W.P.; supervision, G.H.; project administration, Y.S.; funding acquisition, W.P. All authors have read and agreed to the published version of the manuscript.

**Funding:** This work was supported by State Grid Electric Power Company Science and Technology Project: No. 5700-202140381A-0-0-00.

**Data Availability Statement:** The data that support the findings of this study are available from the corresponding author upon reasonable request.

**Conflicts of Interest:** The authors declare no conflict of interest.

## Abbreviations

The following abbreviations are used in this manuscript:

| | |
|---|---|
| 5G | 5th Generation Mobile Communication Network |
| DEP | Detection Error Probability |
| GP | Geometric Program |
| LoS | Line of Sight |
| MDP | Miss Detection Probability |
| NOMA | Non-orthogonal Multiple Access |
| SCA | Successive Convex Approximation |
| SGPA | Successive Geometric Programming Approximation |

| SIC | Successive Interference Cancellation |
|-----|--------------------------------------|
| SP | Signomial Programming |
| UAV | Unmanned Aerial Vehicle |
| SNR | Signal-to-Interference and Noise Ratio |
| FAP | False Alarm Irobability |
| MAP | Missed Alarm Probability |
| KL | Kullback- Leibler |
| CVX | Convex |
| CCS | Covert Communication System |
| SGP | Symbolic Geometric Programming |
| IRS | Intelligent Reflecting Surface |
| LTMB | Length of Transmission Message Block |

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
