# Peer review of "On Joint Optimization of UAV-Assisted Covert Communication Systems with NOMA for Hydropower Internet of Things"

_drones, doi:10.3390/drones7100610_

Round 1
Reviewer 1 Report
This paper jointly optimizes the transmit power, the altitude and the trajectory of the UAV to maximize the covert rate. This paper is well organized and easy to follow. The figures and tables are readily understandable, and they support the text. The detailed comments for revisions are listed as follows:
1. In the Introduction, the authors introduce NOMA and UAV-assisted communications. However, there are many related work on UAV-assisted NOMA communications are not referenced. The authors should put more effort into the related work and supplement the introduction. For instance, the following papers have addressed the UAV placement and NOMA power allocation problem;
Y. Yin, M. Liu, G. Gui, H. Gacanin and H. Sari, "Minimizing Delay for MIMO-NOMA Resource Allocation in UAV-Assisted Caching Networks," in IEEE Transactions on Vehicular Technology, vol. 72, no. 4, pp. 4728-4732, April 2023, doi: 10.1109/TVT.2022.3225058.
I suggest the authors read this paper and explain the contribution and innovation compared to this paper:
X. Jiang et al., "Covert Communication in UAV-Assisted Air-Ground Networks," in IEEE Wireless Communications, vol. 28, no. 4, pp. 190-197, August 2021, doi: 10.1109/MWC.001.2000454.
2. In the second paragraph of System Model, the authors adopt a power allocation method for NOMA users, but they did not describe the specific execution steps for power allocation. The authors are suggested to rearrange the paragraph arrangement of this section.
3. In air-to-ground communication, there is a high probability of line-of-sight links, but the probability is not 100%. The authors should explain the applied channel model between the UAV and the ground user.
4. Eq. 5 is the signal to interference plus noise ratio of U1, instead of signal-to-noise ratio.
5. On line 185, the uppercase T should be changed to lowercase. The authors should review the entire text to avoid such errors.
6. The axis of Figure 3 is wrong, since the iteration numbers should be integers.
In the second paragraph of System Model, the authors adopt a power allocation method for NOMA users, but they did not describe the specific execution steps for power allocation. The authors are suggested to rearrange the paragraph arrangement of this section.
On line 185, the uppercase T should be changed to lowercase. The authors should review the entire text to avoid such errors.
Author Response
Dear reviewer,
We appreciate your comments, and we have finished the response to your comments. Please see the attached file. Thanks!
Best Regards
All Authors

Reviewer 2 Report
The authors proposed a UAV-assisted covert communication system (CCS), where a UAV acts as the base station to provide communication service to ground terminals monitored by malicious users. To improve access effectiveness, this paper adopted non-Orthogonal multiple access (NOMA) for intelligent terminals to access the hydropower IoT. Both the presentation and the contribution are OK. Some comments are given as below.
(1) Some related works should be discussed and surveyed. There are also some related and advanced works in recent years.
(2) The algorithm statement should be provided.
(3) There are some English writing and typos. Also, there have format problems.
There are some English writing and typos. Also, there have format problems.
Author Response
Dear reviewer,
We appreciate your comments, and we have finished the response to your comments. Please see the attached file, named drone_response_2.pdf. Thanks!
Best Regards

Reviewer 3 Report
Dear Authors, Please carefully check and solve all observations below as formulated and sent to the editors of the Drones journal, to which you submitted your paper:"Dear Editors,
This paper (id drones-2531796, entitled “On Joint Optimization of UAV-assisted Covert Communication Systems with NOMA for Hydropower Internet of Things”) proposes a UAV-assisted covert communication system (CCS), where a UAV acts as the base station to provide communication service to ground terminals monitored by malicious users. the authors claim that the simulation results show that the proposed CCSm can achieve a positive covert rate and the developed algorithm can fast convergence in comparison with traditional optimization algorithms
After reading this manuscript, I think I found some issues the authors should deal with. I will start with the format ones. Then I will continue with those related to the paper’s content/substance:
-
In terms of English language and style issues, https://app.grammarly.com on default settings (American English, Set Goals: Audience=Knowledgeable, Formality=Neutral, Domain=General) detected only for the text block resulting from the concatenation of Title+Abstract+Conclusion(s): 0 (zero) correctness issues / critical alerts but 21 more complex ones / advanced suggestions. The resulting Grammarly overall/total score as reported by this online tool was just 83 (left edge of Good i.e >=80 and <=85, but still not Very Good/Excellent i.e. >=90) out of 100 (max) for this three-component sample above. Therefore, I suggest a minimum revision of the English language and style for the entire article using Grammarly or another specialized tool;
-
The paper must follow the entire set of instructions of the journal;
-
All references to equations/formulas must be explicitly and precisely formulated in the main text - e.g., “... (see eq./formula N).”, and not using “by”, “we have”, or similar;
-
The authors must avoid ending some sections/subsections with formulas, figures, tables, or other components (e.g., eq.14 just before section 3, etc. and no explanatory text after). The authors are required to check the entire manuscript for similar issues;
-
The authors must additionally ensure that all figures have the required resolution (Halftones should have a minimum resolution of 300 dpi & Combination artwork should have a minimum resolution of 600 dpi, according to the Journal’s instructions at the link above). Some of the figures are problematic in these terms (e.g., Fig.1 - easily noticeable large pixels on edges when zooming). The image compression options should be also checked when exporting to the .pdf format (only if this task was performed by the authors);
-
The legend texts/titles of some of the figures are too large (more than one line). The authors should move part of these text blocks in the main text of the manuscript near the first reference to them. The authors are also required to check the entire manuscript for such issues;
-
There is a considerable number of figures (I counted 7) in this manuscript. Some of them which are considered by the authors not essential for understanding the main flow of ideas in the manuscript must be moved to the Appendix section (and correspondingly renumbered and referred - e.g. Fig.A1..An for Appendix A, and so on). If this section is not existing, the authors must create one;
-
The authors are also required to include more explanations and precise details about the standard view of the accuracy values (page 9, line 239, >=70% and <80%-fair models; >=80% and <90%-good models; >=90%-very good/excellent models) and their application here. They should provide more references to scientific papers where this topic is considered and the accuracy intervals are precisely defined;
-
The final list of 28 references is not enough. Given this interdisciplinary paper at the intersection of many consecrated fields, the existing list of references indicates as very probable that many important related contributions in journal papers have left (intentionally or not) not cited in the Related Work/Literature Review section;
-
Moreover, the section dedicated to the interpretation of the results (Results and Discussions, or similar) needs a lot more development and cited references to similar/comparable/different results (already published articles in highly rated scientific journals);
-
The authors must understand that replicability as a fundamental principle in science (https://doi.org/10.1007/s10516-021-09610-2 https://doi.org/10.1038/nature.2016.20504 ) starts with data and it is not a fad but a necessity. Therefore, they should insert in the Data Availability Statement section at the end of the manuscript all precise links to all data providers’ / own databases or datasets or even just simple input files (boundary data included) used when testing the entire approach. They must be aware that no barrier (including cheap formalisms such as “reasonable request”) should stand against science and research. Of course, that data must be anonymized first;
-
Following this scientific principle above, the authors should provide full details about the software (including the precise name of the provider and version number of all the tools/apps) and also complete details about the hardware they used to test their approach and obtain the results presented in this manuscript;
-
Following the same replicability principle, if some own algorithms have been used (e.g. subsection 3.5), the authors must precisely identify them in their own GitHub (or similar) repository section. If not existing, the authors must create one for the entire project corresponding to this manuscript. Otherwise, the authors must clearly specify that no custom algorithms have been used;
-
The List of Abbreviations at the end of this paper is needed (distinct section just after the Conclusions);
-
The same for the Limitations corresponding to the research approach used (distinct section just before the Conclusions).
Thank you for the opportunity to read and check this manuscript!"
-
In terms of English language and style issues, https://app.grammarly.com on default settings (American English, Set Goals: Audience=Knowledgeable, Formality=Neutral, Domain=General) detected only for the text block resulting from the concatenation of Title+Abstract+Conclusion(s): 0 (zero) correctness issues / critical alerts but 21 more complex ones / advanced suggestions. The resulting Grammarly overall/total score as reported by this online tool was just 83 (left edge of Good i.e >=80 and <=85, but still not Very Good/Excellent i.e. >=90) out of 100 (max) for this three-component sample above. Therefore, I suggest a minimum revision of the English language and style for the entire article using Grammarly or another specialized tool;
Author Response
Dear reviewer,
We appreciate your comments, and we have finished the response to your comments. Please see the attached file. Thanks!
Best Regards

Reviewer 4 Report
There is limited literature supporting the work especially in the introduction, and the motivation for Hydropower Internet of Things in this paper. I had serious concern with the authors claiming that the work is applicable to Hydropower Internet of Things, and claiming that they are contributing in that area.
It seems that Hydropower IoT is just used to market the work and make it look like that is the novelty in the work. The Hydropower IoT system described in the paper is no different from the known wireless communication where drones are used as base station for terminal.
How is hydropower IoT different to any other IoT or traditional communication?
There is no justification for adding OFDM in the proposed Hydropower IoT system apart from saying "traditionally OFDM has been used to connect increasing number of terminals"
The paper is talking about Hydropower IoT as a motivation for utilizing a UAV as a base station, and for using NOMA. However, I fail to see why the authors mention Hydropower IoT because the system they are talking about is a normal system of having a UAV as a base station to communicate with terminals using NOMA. . There is nothing new with that system. The authors make an attempt to include Hydropower IoT to justify the work, however, there is no description of the Hydropower IoT system to show why it is different from know systems using drones as base station.
The English is readable, but it can be improved by taking it to an editor to make some sentences more understandable
Author Response
Dear reviewer,
We appreciate your comments, and we have finished the response to your comments. Please see the attached file. Thanks!
Best Regard
The authors

Round 2
Reviewer 3 Report
Dear Authors,
You performed many improvements.
I think the paper is now closer to publication.
I wish you all the best!
Minor checks required.
Author Response
Dear Reviewer,
We appreciate your positive comments. Have a great day.
Sincerely
The Authors!